# Revisiting Assessment of Computational Methods for Hi-C Data Analysis

**DOI:** 10.3390/ijms241813814

**Published:** 2023-09-07

**Authors:** Jing Yang, Xingxing Zhu, Rui Wang, Mingzhou Li, Qianzi Tang

**Affiliations:** 1Livestock and Poultry Multi-Omics Key Laboratory of Ministry of Agriculture and Rural Affairs, College of Animal Science and Technology, Sichuan Agricultural University, Chengdu 611130, China; 15283781283@163.com (J.Y.); m13228149385@163.com (X.Z.); ryan.wang.sicau@gmail.com (R.W.); 2Animal Breeding and Genetics Key Laboratory of Sichuan Province, Sichuan Animal Science Academy, Chengdu 610066, China

**Keywords:** topologically associating domains, chromatin interactions, promoter–enhancer interactions

## Abstract

The performances of algorithms for Hi-C data preprocessing, the identification of topologically associating domains, and the detection of chromatin interactions and promoter–enhancer interactions have been mostly evaluated using semi-quantitative or synthetic data approaches, without utilizing the most recent methods, since 2017. In this study, we comprehensively evaluated 24 popular state-of-the-art methods for the complete end-to-end pipeline of Hi-C data analysis, using manually curated or experimentally validated benchmark datasets, including a CRISPR dataset for promoter–enhancer interaction validation. Our results indicate that, although no single method exhibited superior performance in all situations, HiC-Pro, DomainCaller, and Fit-Hi-C2 showed relatively balanced performances of most evaluation metrics for preprocessing, topologically associating domain identification, and chromatin interaction/promoter–enhancer interaction detection, respectively. The comprehensive comparison presented in this manuscript provides a reference for researchers to choose Hi-C analysis tools that best suit their needs.

## 1. Introduction

Genomic DNA, of over 6 inches in length, is folded and packed inside of the nucleus, which is less than 10 µm in diameter; this forms a complicated spatial organization. This three-dimensional structure of chromatin is believed to play a critical role in transcriptional and functional regulation, in both physiological or pathological conditions. Hi-C, a method based on chromosome conformation capture sequencing [1], combines proximity-based DNA ligation and high-throughput sequencing to measure the spatial proximity of two genomic loci. Hi-C analysis revealed hierarchical 3D chromatin structures, including territories, compartments, topologically associating domains (TADs) [2], chromatin interactions (CIs) [3], and promoter–enhancer interactions (PEIs). In particular, TADs are genomic regions with significantly more intra-domain interactions than inter-domain ones; CIs are spatial contacts between pairs of loci that are far from each other on the linear DNA sequence; if these two loci each reside in promoter and enhancer regions, their contact is considered a PEI.

Hi-C generates millions of read pairs that can be further processed to produce genome-wide two-dimensional contact maps, with the number of read pairs residing in each 2D bin connecting one pair of linear DNA loci representing the spatial proximity of the two loci [4,5,6]. The large quantity and inherent bias of the data require specialized algorithms and efficient software. Different bioinformatic tools have been developed to pre-process read pairs (quality control, alignment, and filtering), bin valid contacts into 2D interaction maps, remove bias from the maps, detect TADs, and identify interactions [6,7]. Assessment and comparison of the performances of mainstream algorithms and computational methods are critical, because distinct tools produce irreproducible results with non-trivial differences.

Dali et al. [8] compared the output of seven different TAD prediction tools on two published Hi-C datasets, found that the number, size, and other biological characteristics of TADs, as predicted using different tools, varied greatly, and suggested that next generations of TAD prediction tools should relax these assumptions regarding the overlap, nesting, and size of TAD structure, so as to better capture the full range of TADs. But this research did not include the latest TAD prediction tools. Forcato et al. [9] quantitatively compared the performances of 13 tools to analyze Hi-C data. The comparison showed that the performances of the tools for identifying chromatin interactions were significantly different, and the results of TAD detection tools were more comparable. Unfortunately, there is no complete assessment for the performances of the tools for identifying PEIs. Zufferey [10] tested and compared 22 computational methods to identify TADs across 20 different conditions and further confirmed that TADs are hierarchically organized domains, but they did not validate their findings with manually curated TADs. Another related work compared capture Hi-C analytical pipelines [11]. They found that the most significant difference among these tools was the number of CIs identified, and the optimal pipeline depends on the project-specific tolerance level of false-positive and false-negative CIs. In summary, previous studies that evaluated Hi-C data analysis methods either conducted evaluations using semi-quantitative or synthetic data approaches [8,9,10], or were outdated and did not include the most recently developed tools.

Here, we comprehensively evaluated and compared the performances of 24 different mainstream Hi-C data analysis methods for data preprocessing, TAD detection, and CI/PEI identification using experimental and manually curated benchmark data. This study is the first large-scale assessment of Hi-C data analysis methods since 2017, complements previous related studies, and guides users to determine the best tools for TAD, CI and PEI detection.

## 2. Results

### 2.1. Data Processing

Different methods preprocess Hi-C data using distinct alignment and filtering strategies (Table 1). Methods using bwa distiller-nf (version 0.3.3) and Juicer (version 1.5.6) [12]) aligned an average of 90.7% read pairs, whereas only an average of 75.0% were mapped using Bowtie2 (HiC-Pro (version 2.11.4) [13], HiCExplorer (version 3.4.1) [14], and HOMER (version 4.11) [15]). The alignment of the method using GEM TADbit (version 1.0) [16]) was 80.67%. After filtering, distiller-nf retained the largest number of aligned reads.

The reproducibility of matrices generated using biological or technical replicates could serve as an evaluation index for assessing the performances of preprocessing approaches. We used HiCRep [17], which considers the unique characteristics of Hi-C data, to measure matrix reproducibility. HiC-Pro exhibited the highest reproducibility for both raw and normalized matrices across replicates, whereas distiller-nf showed only comparably high reproducibility for the normalized matrices (Figure 1A). 

As distiller-nf is the preprocessing method recommended by the 4D Genome Project, we assessed the similarities between distiller-nf and other tools. HiC-Pro and Juicer showed the highest similarity to distiller-nf based on HiCRep (Figure 1B). Using HiC-Spector, a reproducibility metric between two Hi-C interaction matrices, we observed that HiC-Pro was the most similar to distiller-nf, followed by Juicer (Figure 1B). We further evaluated the similarities between distinct tools in a pairwise manner and observed relatively consistent results with HiCRep and HiC-Spector, in which Juicer and HiC-Pro exhibited the highest similarity (Figure 1C).

### 2.2. Comparison of TAD Identification Tools

To compare the performances of distinct TAD identification tools, we generated contact matrices binned into two resolutions (25 and 50 kb) and derived data from the sampled total reads of three depths (100 M read pairs; 500 M read pairs; and the complete dataset, termed full) (Appendix A); then, we applied the 11 most popular software programs (Appendix A). The TAD size was relatively stable across different sequencing depths, with few exceptions (i.e., Arrowhead (version 1.8.9) [5,12]). However, at the same sequencing depth, TAD size decreased with increasing resolution. The average TAD sizes range from 131 kb, for Armatus (version 2.3) [18] at 100 M sequencing depth and 25 kb resolution, to 1.75 Mb for HiCseg (version 1.1) [19] at full sequencing depth and 50 kb resolution, with the average TAD size for most tools being close to 500 kb (Appendix A). As expected, the total TAD number is the opposite of the TAD size; Armatus and TADtree (publicly available at http://compbio.cs.brown.edu/projects/tadtree/ (accessed on the 17 June 2021)) [20] detect more TADs than other tools, with both tools producing more than 15,000 TADs at 2 kb resolution and more than 8500 TADs at 50 kb resolution (Appendix A). Arrowhead, OnTAD (version 1.3) [21], and TADtree return nested TADs, and other tools generate non-overlapping TADs (Figure 2A).

To directly quantify the similarity of the TADs called with different tools, we calculated the Measures of Concordance (MoCs) [10] of TAD intervals and the Jaccard Index (JI) of individual TAD boundaries between distinct tools at three sequencing depths and two resolutions. The MoCs of TAD intervals were robust to variation in sequencing coverage. At 25 kb resolution, DomainCaller (version 1.0) [2] and InsulationScore (version 1.0.0) [22] exhibited the highest similarity, followed by HiCExplorer (version 3.4.1) [14] and TopDom (version 0.0.2) [23] (Figure 2B). At 50 kb resolution, DomainCaller, HiCExplorer, and TopDom exhibited the highest similarity. Alternatively, HiCDB (publicly available at https://github.com/ChenFengling/HiCDB (accessed on the 21 June 2021)) [24] and Armatus showed the lowest similarities compared with other tools at all levels (Figure 2B). The JI of TAD boundaries showed consistent results, and DomainCaller, HiCExplorer, and TopDom were highly similar at three sequencing depths and two resolutions. In addition, TAD boundaries called at 50 kb resolution showed generally higher similarity between tools than those at 25 kb (Figure 2C).

For each tool, we evaluated the positive predictive value (PPV) between the predicted and manually curated TADs. Tools usually performed much better at predicting TAD boundaries than intervals, with boundary prediction PPVs ranging from 15% to 61%, and TAD interval prediction PPVs rarely exceeding 15%. In addition, accurate boundary detection does not necessarily represent accurate TAD prediction, especially for HiCDB (Figure 2D,E). TAD intervals called with DomainCaller exhibited the highest similarity to the manually curated TADs, whereas HiCDB produced the poorest results (Figure 2D). Armatus, InsulationScore, HiCDB, TopDom, HiCExplorer (version 3.4.1) [14], TADtree, and DomainCaller produced TAD boundary predictions that were generally consistent with manually curated TADs and were robust to variation across sequencing depths (Figure 2E).

We further evaluated the reproducibility of the TADs called with the same tools across varied sequencing depths and resolutions, using the MoCs of TAD intervals and the JIs of individual TAD boundaries. In general, reproducibility is higher across distinct sequencing depths than across different resolutions (Figure 3A,B). With the exception of Arrowhead, whose reproducibility for distinct resolutions is affected by sequencing depth, the other tools showed highly similar reproducibility for distinct resolutions across three sequencing depths (Figure 3A,B). For TAD intervals, HiCseg produced the most reproducible results, whereas, for TAD boundaries, HiCseg and InsulationScore showed the highest stability, across both resolutions and sequencing depths (Figure 3A,B).

TADs were originally defined as genomic intervals with more intra- than inter-interval interactions; therefore, we reasoned that the accuracy of identified TADs could be measured by comparing the intra- and inter-TAD interaction frequencies. We observed that DomainCaller, HiCExplorer [14], InsulationScore, and TopDom showed the largest discrepancies between intra- and inter-TAD interaction frequencies (log_2_[fold change] > 2, *p*-value < 2.2 × 10^−16^) (Figure 3C). In particular, DomainCaller consistently exhibited the highest ratio of intra- to inter-TAD interaction frequencies across three sequencing depths. Tools generating nested TADs, such as Arrowhead, OnTAD, and TADtree, showed the weakest enrichment of intra- compared with inter-TAD interaction frequencies (Figure 3C).

Considering that it is not fair to use this metric for tools returning nested TADs, we only selected the outer-layer TADs for tools returning nested TADs, and the performances of the three tools were significantly improved (Figure 3D). Then, compared with other tools, DomainCaller still showed the greatest discrepancies in interaction frequencies, robust at three sequencing depths (Figure 3D). We also separately compared the three tools that returned nested TADs and removed the outer-layer TADs to obtain the inner-layer TADs. The violin plot showed that Arrowhead had more fold change in intra- and inter-TAD interaction frequencies, followed by TADtree and OnTAD; the results were consistent with the comparison of TADs without hierarchy (Appendix A).

A second feature of TADs, that of the enrichment of CTCF binding at boundaries, was further applied to measure the accuracy of different tools. Most tools exhibited significant enrichment of CTCF peaks around their predicted TAD boundaries; the CTCF peak enrichment of Amatus was more evident at 50 kb resolution compared with 25 kb resolution, and Arrowhead performed better at low sequencing depth and high resolution, while the other tools were robust at all sequencing depths and resolutions. Notably, HiCDB showed the highest enrichment of CTCF peaks across all of the measured sequencing depths and resolutions (Figure 4A). the CTCF peak enrichment in other datasets are shown in Appendix A.

Similarly, we also selected the outer-layer TAD for tools returning nested TADs, to compare whether the TAD boundary was enriched for CTCF binding. The visualization results show that the enrichment peaks using only the outer layer of the TAD boundary are sharper, especially OnTAD, which is second only to the HiCDB tool (Figure 4B and Appendix A). In contrast, CTCF binding enrichment of the inner-layer TAD was essentially unchanged in both datasets (Appendix A). Since the CTCF binding site plays a role in maintaining the structural stability of the TAD, we speculate that the large-sized TAD requires more CTCF binding to maintain structural stability [25].

### 2.3. Identification of PEIs

We used Rao’s [5] K562 dataset to generate contact matrices at 10 kb resolution, then systematically evaluated the performances of distinct CI callers for identifying PEIs, as well as one additional tool, PSYCHIC (publicly available at https://github.com/dhkron/PSYCHIC (accessed on the 2 March 2021)) [26], which was specifically designed for PEI identification (Appendix A). Among the 11 most popular CI callers, PEIs accounted for 15% (HiC-DC+ (version 0.99.13) [27]) to 35% (cLoops (version 0.93) [28]) of identified interactions, with a PEI proportion close to 20% for most tools (Figure 5A). Notably, the number of identified PEIs greatly varied across distinct tools, ranging from 705 for SIP (version 1.6.1) [29] to 131,863 for GOTHiC (version 1.22.0) [30] (Appendix A). Four tools (GOTHiC, PSYCHIC, HOMER (version 4.11) [15], and Fit-Hi-C2 (version 2.0.7) [31]) produced over 120,000 PEIs each, followed by 22,868 using HiC-DC+, with the other tools producing below 4000 (Appendix A). Considering the distances between promoters and enhancers, over 85% of PEIs predicted using HOMER and GOTHiC were located within 80 kb (Figure 5B). Excluding these two tools, PEIs were generally distant from each other; in particular, PEIs identified using tools specifically designed for loop detection, such as cLoops, HiC-DC+, and SIP, were mostly long-range (Figure 5B).

It is known that more PEIs reside intra-TAD than inter-TAD; therefore, we used TADs identified with HiCExplorer (version 3.4.1) [14] at 25 kb resolution with the full dataset to estimate the proportion of PEIs located intra-TAD, inter-TAD, and on TAD boundaries to evaluate PEI calling accuracy. We also included pseudo-PEIs generated by selecting randomly paired promoters and enhancers to match the same distance distribution as the PEIs identified with the distinct tools. cLoops and HiC-DC+ produced the lowest proportion of PEIs within the same TAD, and a substantial number of PEIs identified using cLoops were located on TAD boundaries (Figure 5D). GOTHiC and HOMER generated high proportions of intra-TAD PEIs, but their paired pseudo-PEIs showed similar behavior, which might be explained by the majority of their PEIs being in close proximity (Figure 5D). Aside from those aforementioned tools, PEIs identified with the other tools exhibited a higher intra-TAD proportion than pseudo-PEIs (Figure 5D).

We further applied the JI as a criterion with which to measure the similarity of PEIs identified using distinct tools. The software programs that produced over 120,000 PEIs each (Fit-Hi-C2, GOTHiC, HOMER, and PSYCHIC) showed high similarity, among which GOTHiC and HOMER were the most similar, which was probably attributable to the detection of a large number of short-range PEIs using the two software programs (Figure 5C). As an example with which to facilitate intuitive comprehension of the differences between distinct tools, we showed contact matrices and identified PEIs for a typical genomic region (Figure 5E).

We further evaluated the accuracy of detected PEIs using the ground truth PEIs from the CRISPR dataset [32]. HiC-DC+ showed the highest precision (20%), but with only 7% recall, whereas the precision for the other tools ranged from 5% to 10% (Figure 5F). The highest recall was achieved using GOTHiC (87%) and HOMER (85%); however, their levels of precision were not significantly higher than that of the pseudo-PEIs (Figure 5F). Notably, Fit-Hi-C2 produced relatively balanced recall (46%) and precision (6%), and both of these statistics were higher than those of the pseudo-PEIs (Figure 5F). To minimize confusion, tools with both precision and recall equal to 0 were not shown in Figure 5F.

Putative enhancers interacting with promoters tend to exhibit higher conservation levels and enrichment of markers for active transcription (H3K27ac), open chromatin (DNase I hypersensitive sites), and insulation (CTCF); therefore, we applied these four criteria for the evaluation of PEI accuracy (Appendix A). The pseudo-enhancers were generated by sampling 10 kb genomic regions with similar distances from the promoters compared with those of putative enhancers. Fit-Hi-C2, GOTHiC, and HOMER showed the highest levels of conservation (Figure 5G), as well as enrichment of H3K27ac peaks (Figure 5H) and DNase I hypersensitive sites (Figure 5H); the enrichment of conservation values for each method are presented in separate figures in Appendix A; alternatively, cLoops, HiCCUPS (version 1.22.01) [5,12], and SIP, which were specifically designed for loop detection, achieved the best results using CTCF peaks as a criterion (Figure 5H).

### 2.4. Identification of CIs

The contact matrices for CI identification were generated using Rao’s [5] GM12878 dataset at 5 kb resolution. We comprehensively evaluated the performances of the 10 most popular software programs for CI identification (Appendix A). To reduce the impact of numerous very proximal interactions on final evaluation using certain methods, we only retained interactions over 25 kb. The median distance in all software programs was less than 1 Mb. Most of the interaction points called with GOTHiC (version 1.22.0) [30] and HOMER (version 4.11) [15] were short-range (Figure 6A). Fit-Hi-C2 (version 2.0.7) [31] called interactions that were mostly mid-range, with an average of ~100 kb (Figure 6A). cLoops (version 0.93) [28] identified relatively long-distance interactions, followed by HiC-DC+ (version 0.99.13) [27] and HiCExplorer (version 3.4.1) [14] (Figure 6A). The discrepancies in the numbers of interactions and distances between the interacting points identified using the distinct tools were obvious when visualizing the contact matrices (Figure 6D).

Considering all methods, and the GM12878 data at 5 kb resolution, 6–18% of all detected interactions were classified as PEIs (Figure 6B). At this resolution, GOTHiC, Fit-Hi-C2, and HOMER called both the highest proportion and absolute number of PEIs; in contrast, cLoops identified the lowest percentage of PEIs.

We assessed the power to recall validated CIs interactions (Figure 6C). Except for Fit-Hi-C2 (16%), HOMER (9.8%), GOTHiC (9.5%), and HiC-DC+ (8.1%), the proportions of true-positive interactions recovered using other tools were quite low (0.4–3.1%) (Figure 6C). Considering both the number and proportion of recalled true-positive interactions, Fit-HiC-2 exhibited relatively balanced performance, with the highest sensitivity and number of recalled true positives (Figure 6C). In contrast, cLoops recalled only 0.4% of true positives with numerous identified interactions (Figure 6C).

## 3. Discussion

The performances of algorithms for preprocessing and identification of TADs, CIs, and PEIs from Hi-C data have been, in most cases, compared using semi-quantitative or synthetic data approaches [8,10]. Dali et al. [9] generated manually curated true-positive TADs, but their work did not include the most recent methods, such as HiCDB (publicly available at https://github.com/ChenFengling/HiCDB (accessed on the 21 June 2021)) [23], HiCExplorer (version 3.4.1) [12], InsulationScore (version 1.0.0) [21], and OnTAD (version 1.3) [20]. In this study, we comprehensively evaluated popular state-of-the-art methods for the complete end-to-end pipeline of Hi-C data analysis using manually curated or experimentally validated true-positive datasets, including a CRISPR dataset [26] for PEI validation.

The TADs identified using distinct tools vary greatly. Assessment based on the analysis of TAD intervals and TAD boundaries showed that no single tool could perfectly identify all TADs. Using manually curated TADs as benchmarks, we observed that, even though predicted TAD boundaries exhibited relatively high accuracy, TAD intervals demonstrated only moderate accuracy, which is consistent with the results of Dali et al. [8]. This might be because many tools do not detect nested TADs, while manually curated TADs contain many overlapping TADs, often nested, but sometimes not. Considering all evaluation metrices, Arrowhead (version 1.8.9) [5,33] seems to be sensitive to sequencing depth, as it works better at lower sequencing depth. Although DomainCaller (version 1.0) [2] does not rank first for every evaluation metric, it shows the greatest discrepancies between intra- and inter-TAD interaction frequencies (log_2_[fold change] > 2.3, *p*-value < 2.2 × 10^−16^), and ranked the highest for other evaluation metrics. In addition, the sizes and number of TADs identified with DomainCaller are balanced, indicating good comprehensive performance.

Instead of using the strategy adopted by Forcato et al. [8], in which CIs were identified from each biological or technical replicate, we only generated CIs and the derived PEIs from pooled datasets that contained sufficient contacts for reliable detection of high-resolution 3D chromatin structures. The identified PEI set showed significantly better performance than the random PEI set for numerous evaluation metrics, indicating that our analysis results are reliable. We observed that GOTHiC (version 1.22.0) [28] and HOMER (version 4.11) [13] tend to call short-range PEIs and interactions. This characteristic explained that almost all of their identified PEIs/CIs were located within TADs, and that they exhibited relatively high recall using the CRISPR data, which mainly detects short-range PEIs as well, as a benchmark. Fit-Hi-C2 (version 2.0.7) [29], designed as a mid-range interaction caller, also produced a relatively balanced performance of evaluation metrics. Considering the short-range limitation of GOTHiC and HOMER, we recommend Fit-Hi-C2 for PEI/CI identification.

In summary, our results indicate that, although no single method exhibits superior performance in all situations, TAD callers are methodologically more stable than interaction callers. From a comprehensive perspective, among TAD callers, DomainCaller showed a relatively balanced performance for most evaluation metrics. For CI callers and derived PEI callers, Fit-Hi-C2 recalled the highest proportion of validated CIs interactions and exhibited a balanced performance for most evaluation metrics for PEI identification.

## 4. Materials and Methods

### 4.1. Input Data

GM12878 (Experiments HIC001 to HIC029) and K562 (Experiments HIC069 to HIC074) Hi-C data by Rao et al. [5] were downloaded from GEO GSE63525. The downloaded SRA files were converted to FASTQ using the SRA Toolkit, and they were further subjected to distinct data preprocessing tools. Here, GM12878 Hi-C data were utilized for TAD and CI identification, whereas K562 Hi-C data were applied for PEI identification (Table 2).

### 4.2. Methods for Data Preprocessing

The tools for aligning Hi-C data, pairing the reads, processing chimeras, and merging and sorting the reads to filter out PCR duplicates involved three main algorithms, namely: the Burrows–Wheeler Aligner (BWA) [34], Bowtie2 [35], and the Genome Multitool (GEM) [33]. Juicer (version 1.5.6) [12] was applied to bin the contacts into matrices for the downstream analyses of most tools, except for distiller-nf (version 0.3.3) and HiCExplorer (version 3.4.1) [14], which rely on cooler (version 0.9.1) [36] to obtain raw matrices. The raw matrices were further normalized using KR or iterative correction and the eigenvector decomposition (ICE) algorithm [37].

#### 4.2.1. Tool Usage

We used a total of 6 tools; please see the Appendix A for details.

#### 4.2.2. The Reproducibility of Hi-C Interaction Matrix

To assess the reproducibility of Hi-C contact matrices, we considered 2 metrics: HiCRep [17] and HiC-spector [38]. We used the 3DChromatin_ReplicateQC (version 1.0.1) [39] for implementations of the two metrics.

HiCRep [17] is a framework for assessing the reproducibility of Hi-C data. It minimizes the effects of noise and biases by smoothing the Hi-C matrices, and addresses the distance-dependent effect by stratifying the Hi-C data based on genomic distances. It further uses a stratum-adjusted correlation coefficient as a measure of Hi-C data reproducibility. The value ranges from −1 to 1, and can be used to estimate the degree of reproducibility.

HiC-spector [38] can also calculate the reproducibility metrics between two Hi-C interaction matrices. It introduces a novel metric for quantifying the reproducibility of the Hi-C contact maps using spectral decomposition, producing repeatability score *Q,* which ranges from 0 to 1. This metric is successful in separating the contact maps of Hi-C data among biological replicates, pseudo-replications, and samples from different cell types.

### 4.3. Methods for the Analysis of TADs

For TAD identification, paired reads were mapped to the human genome hg19 using distiller-nf (version 0.3.3). Lower-depth datasets were obtained by down-sampling the GM12878 dataset [5] to ∼100 M and 500 M paired reads. Raw and ICE-normalized contact matrices were generated, using the HiCExplorer (version 3.4.1) [14] at 25 kb and 50 kb resolutions, and were used as inputs for the TAD detection tools. The sequencing depth and binning resolutions for TAD identification were chosen based on the methods of Dali et al. [8].

#### 4.3.1. Tool Usage

In total, we assessed 11 TAD callers (Figure 7); please see the Appendix A for details of each method.

#### 4.3.2. The Concordance of TAD Intervals

The concordance between TAD intervals was measured using the Measure of Concordance (MoC), a metric for similarity measurements between pairs of clusters [10]. As the base pairs in the TAD intervals can be treated as elements in clusters, TAD intervals can be treated as clusters. The MoC is defined as follows, where *P* and *Q* are two sets of TAD intervals, including $N_{P}$ and $N_{Q}$ TAD intervals. $P_{i}$ and $Q_{j}$ are two separate TADs in *P* and *Q*, with sizes $\left\|P_{i}\right\|$ and $\left\|Q_{j}\right\|$, respectively. $\left\|F_{i,j}\right\|$ is the overlap between $P_{i}$ and $Q_{j}$. The MoC takes the value 0 when there is independence between *P* and *Q*, and 1 when *P* = *Q*.
(1)$$\mathrm{MoC}\left(P,Q\right)=\left\{\begin{array}{c} 1:\mathrm{if}\;N_{P}=N_{Q}\\ \mathrm{otherwise}:\frac{1}{\left(\sqrt{N_{P}N_{Q}}-1\right)}\left(\sum_{i=1}^{N_{P}}\sum_{j=1}^{N_{Q}}\frac{{\left|\left|F_{i,j}\right|\right|}^{2}}{\left||P_{i}||\;||Q_{j}|\right|}-1\right) \end{array}\right.,$$

#### 4.3.3. The Concordance of TAD Boundaries

The similarity between TAD boundaries was assessed using the Jaccard Index (JI), where JI is defined as the size of the intersection divided by the size of the union between two finite sets.
(2)$$\mathrm{JI}\left(A,B\right)=\frac{\left|A⋂B\right|}{\left|A⋃B\right|},\;$$

*A* and *B* represent the sets of two TAD boundaries. We took 1 kb from both the left and right sides of TAD intervals as the set of TAD boundaries, and then used the bedtools jaccard [40] function to calculate the JI. As a result, the final statistic ranges from 0 to 1, where 0 represents no overlap and 1 represents complete overlap.

#### 4.3.4. Manually Curated TADs

Dali et al. [8] used Adobe Illustrator to manually trace visually identifiable TAD regions from GM12878 and hESC interaction maps at full sequencing depth and 50 kb resolution, and manually curated TADs were required to meet the following two conditions: (i) sharp visual contrast between intra- and inter-TAD interaction frequencies, over the entire TAD region; and (ii) minimum size of 250 kb. They randomly selected the 40-45 Mb region of 10 chromosomes (chr2, chr3, chr4, chr5, chr6, chr7, chr12, chr18, chr20, and chr22) for manual annotation. Finally, the sizes of manually curated TADs ranged from a few hundred kilobases to several megabases, with an average size of approximately 650 kb. In addition, manually curated TADs contained many overlapping TADs, often nested, but sometimes not.

### 4.4. Methods for the Analysis of CIs and PEIs

Similar to data preprocessing for TAD identification, distiller-nf was applied to align the sequences of the K562 Hi-C reads [5], parse .sam alignment, form files with Hi-C pairs, and filter PCR duplicates. The aligned, paired, and duplicate-removed reads were retained for downstream analyses of CI and PEI identification.

#### 4.4.1. Tool Usage

We assessed 11 tools for CI and PEI identification (PSYCHIC [26] was only used for the assessment of PEI identification) (Figure 7); please see the Appendix A for details of each method.

#### 4.4.2. PEIs and Random PEIs

There are two anchors for CIs identified using tools. If the transcription start site is located in one anchor, and the other anchor does not contain a transcription start site, the CI is considered to be a PEI. We further simulated random PEIs as background. The number and the distances between two anchors of random PEIs are based on PEIs. Firstly, the distances between two anchors of PEIs were sorted, removing the nearest and furthest top 1% of extreme cases. Then, the distances were divided into 15 groups in ascending order, with an equal number of PEIs in each group. In addition, we took all of the anchors of PEIs as a set and randomly sampled two anchors from the set. If the distance of two random anchors fell within the distance range of a certain group and did not overlap with a PEI, it was retained. The process continued until the number of random PEIs was equal to the number of PEIs for each distance group, that is, random PEIs show almost the same distance distribution as PEIs.

#### 4.4.3. CRISPR Dataset for PEI Validation

More than 3500 potential enhancer–gene connections for 30 genes were tested using CRISPRi-FlowFISH [32], a combination of CRISPRi (a gene interference technique) and FISH (a gene staining technique), which interferes with the nucleotide sequences of candidate enhancers near the target gene, and quantifies the effects of these sequences on target genes. The main principle is that guide RNA (gRNA) can guide KRAB-dCas9 to bind to a specific nucleotide sequence and inhibit its expression. KRAB-dCas9 has been shown to inhibit many promoters and enhancers and affect candidate regulatory elements within 200-500 base pairs near gRNA. Ground truth PEIs were derived from the dataset by excluding PEIs with promoters and enhancers residing in the same bin, and retaining those with log-transformed[fold change] < 0 and significance = TRUE. For a single enhancer spanning two adjacent bins, we evaluated the 10 kb bin where the midpoint of the enhancer was located.

#### 4.4.4. Validated CIs Interactions

Sanyal et al. [41] developed an in-house ‘5C peak calling’ algorithm with which to distinguish significant looping interactions from background looping interactions. They called peaks in each 5C biological replicate separately and used the peaks that were shared between replicates as their final list of significant looping interactions. Finally, they obtained 1187 significant looping interactions from the GM12878 dataset, which were used as ground truth for our CI assessment.

## 5. Conclusions

In summary, although distinct methods exhibited different performances, based on varied evaluation criteria, HiC-Pro, DomainCaller and Fit-Hi-C2 showed relatively balanced metrics.

HiC-Pro did not show the largest number of valid mapped read pairs, but its results were the most similar to the 4D Genome Project’s recommended preprocessing method, named distiller-nf (Appendix A). HiC-Pro also exhibited the highest reproducibility across replicates, both in the original and normalized matrices, while distiller-nf exhibited high reproducibility only in the normalized matrices (Appendix A). Therefore, we recommended HiC-Pro for data pre-processing.

Regarding TAD detection, DomainCaller ranked in the top three based on the PPVs of TAD intervals, the reproducibility of TAD intervals and TAD boundaries across two resolutions, the fold change of intra- and inter-TAD interaction frequencies, and the enrichment of CTCF binding (Appendix A). Based on other metrics, DomainCaller was also among the tools ranking high in performance, indicating its high accuracy and stability (Appendix A).

In terms of performance for CI identification, GOTHiC, Fit-Hi-C2, and HOMER ranked as the top three (Appendix A). For PEI identification, the performances of the three tools mentioned above was also similar, ranking as the top three for precision and recall of PEIs, as well as for the conservation levels and enrichment of markers for active transcription and open chromatin (Appendix A). But GOTHiC and HOMER tend to identify short-range PEIs and CIs, so we believe that Fit-Hi-C2 exhibited balanced performance for most evaluation metrics for PEI and CI identification.

Hi-C data analysis tools are proliferating, and evaluating the performances of these tools will always be necessary. This study is the first large assessment of the tools available since 2017 and thoroughly explores tools for Hi-C data analysis, with the limitation of including only two datasets of human beings. We expect that a new perspective on benchmark studies does not only display a comparison of the results, but also offers users a way to interactively explore, replicate these comparisons, easily run analyses, and even include their own tool among the others.

## Figures and Tables

**Figure 1 ijms-24-13814-f001:**
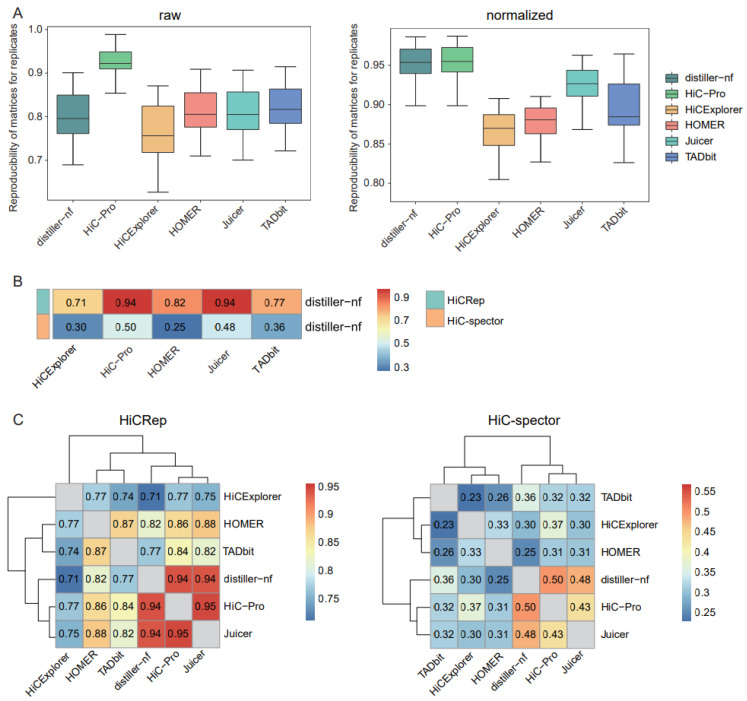
Comparison of matrix reproducibility and similarities. (**A**) Boxplots of the reproducibility of raw and normalized matrices for replicates using HiCRep. (**B**) Heatmap of similarities of normalized Hi-C matrices between distiller-nf and the other five tools, compared using HiCRep and HiC-Spector. (**C**) Heatmaps for similarities of normalized Hi-C matrices between each pair of tools, compared using HiCRep and HiC-Spector.

**Figure 2 ijms-24-13814-f002:**
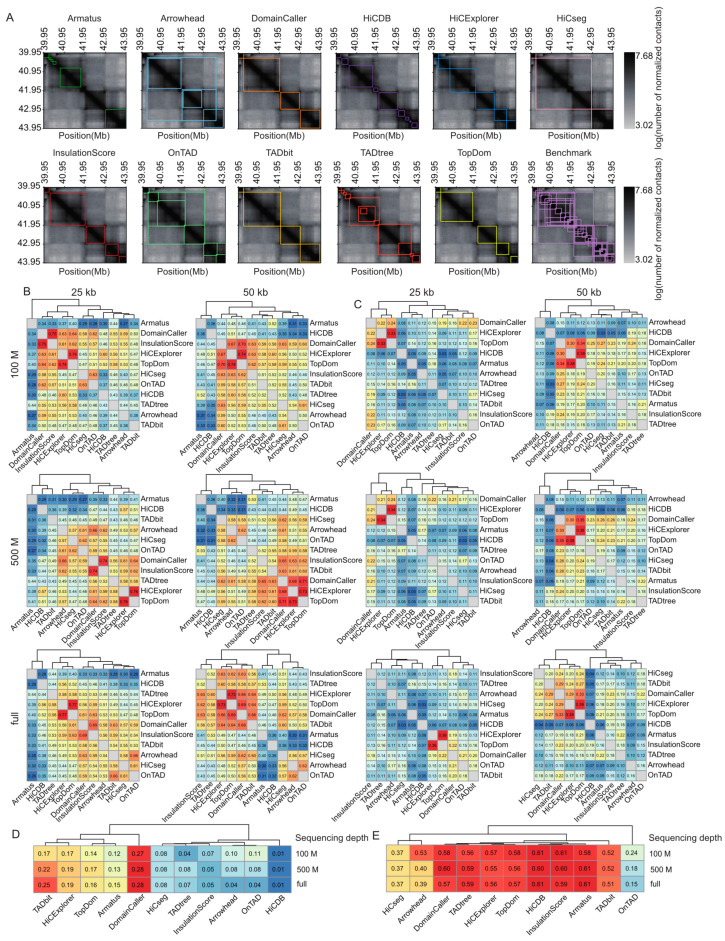
Comparison of TAD identification tools evaluated using Rao’s GM12878 dataset. (**A**) Heatmaps of the contact matrices (chr7:39,950,000-43,950,000) at 50 kb resolution and full sequencing depth. Identified TADs are framed in different colors for various tools. (**B**) Heatmaps of the Measure of Concordance for the concordance of TAD intervals between distinct tools at three sequencing depths and two resolutions. The color gradient from blue to red represents low similarity to high similarity. (**C**) Heatmaps of the Jaccard Index for the concordance of TAD boundaries between distinct tools at three sequencing depths and two resolutions. (**D**) Heatmap of positive predictive values for the concordance of TAD intervals between identified TADs and manually curated TADs at three sequencing depths and 50 kb resolution. (**E**) Heatmap of positive predictive values for the concordance of TAD boundaries between identified TADs and manually curated TADs at three sequencing depths and 50 kb resolution.

**Figure 3 ijms-24-13814-f003:**
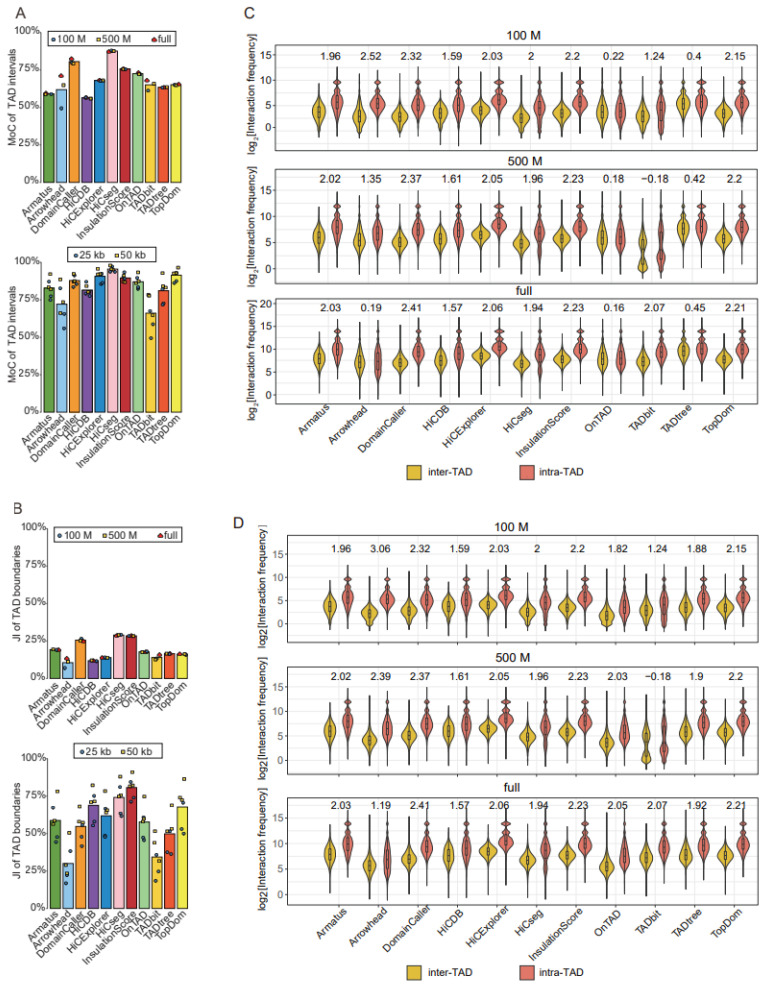
Comparison of TAD identification tools evaluated using Rao’s GM12878 dataset. (**A**) Barplots for stability of TAD intervals compared across two resolutions (25 kb and 50 kb) at three sequencing depths (100 M, 500 M, and full) (upper panel), and across three sequencing depths (100 M, 500 M, and full) at two resolutions (25 kb and 50 kb) (lower panel). (**B**) Barplots for stability of TAD boundaries compared across two resolutions (25 kb and 50 kb) at three sequencing depths (100 M, 500 M, and full) (upper panel), and across three sequencing depths (100 M, 500 M, and full) at two resolutions (25 kb and 50 kb) (lower panel). (**C**) Violin plots of the intra-TAD interaction frequency versus inter-TAD interaction frequency for different tools at 50 kb resolution across three sequencing depths. Inter-TAD interaction frequency corresponds to pairs of bins that are located in adjacent TADs. Intra-TAD interaction frequency corresponds to pairs of bins inside the TADs. The number above each pair of violins represents log_2_[fold change], the Wilcoxon test was performed for all comparisons, and the *p*-values were less than 2.2 × 10^−16^. (**D**) Violin plots of the intra-TAD interaction frequency versus inter-TAD interaction frequency for different tools at 50 kb resolution across three sequencing depths. Arrowhead, OnTAD, and TADtree use the outer layer of the detected nested TAD.

**Figure 4 ijms-24-13814-f004:**
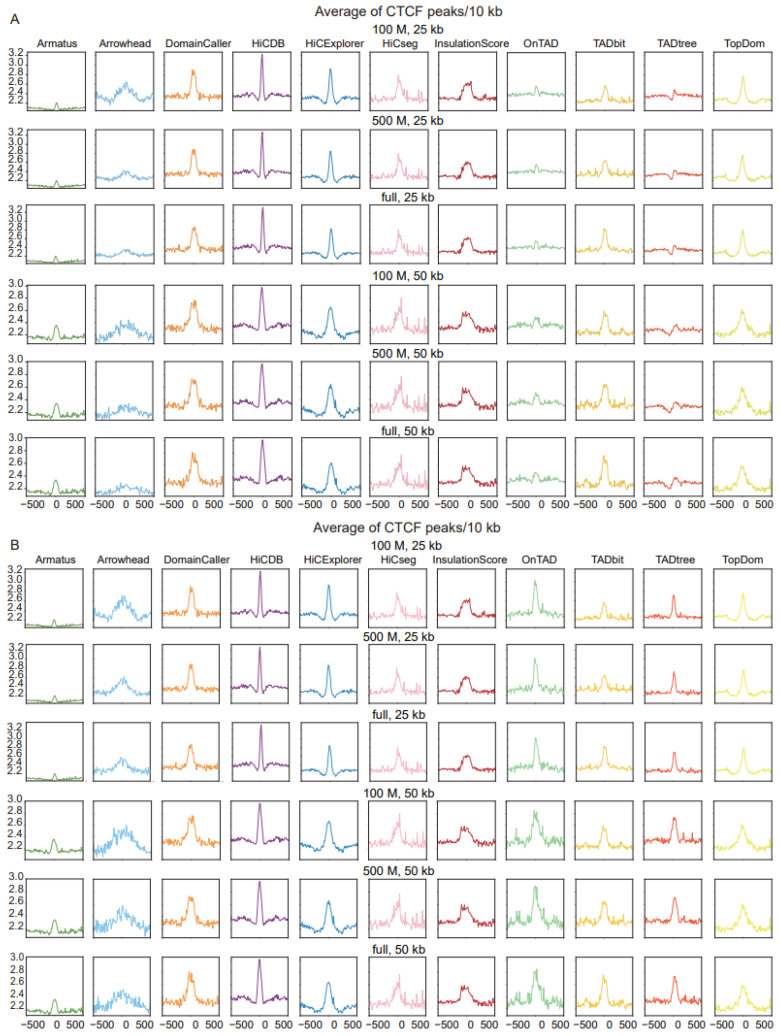
Comparison of TAD identification tools, evaluated using Rao’s GM12878 dataset and CTCF binding dataset (GSE30263). (**A**) Enrichment of CTCF binding in a window of 1 Mb (±500 kb) around the predicted TAD boundaries at three levels of sequencing depths and two resolution levels. (**B**) Enrichment of CTCF binding in a window of 1 Mb (±500 kb) around the predicted outer-layer TAD boundaries at three levels of sequencing depths and two resolution levels.

**Figure 5 ijms-24-13814-f005:**
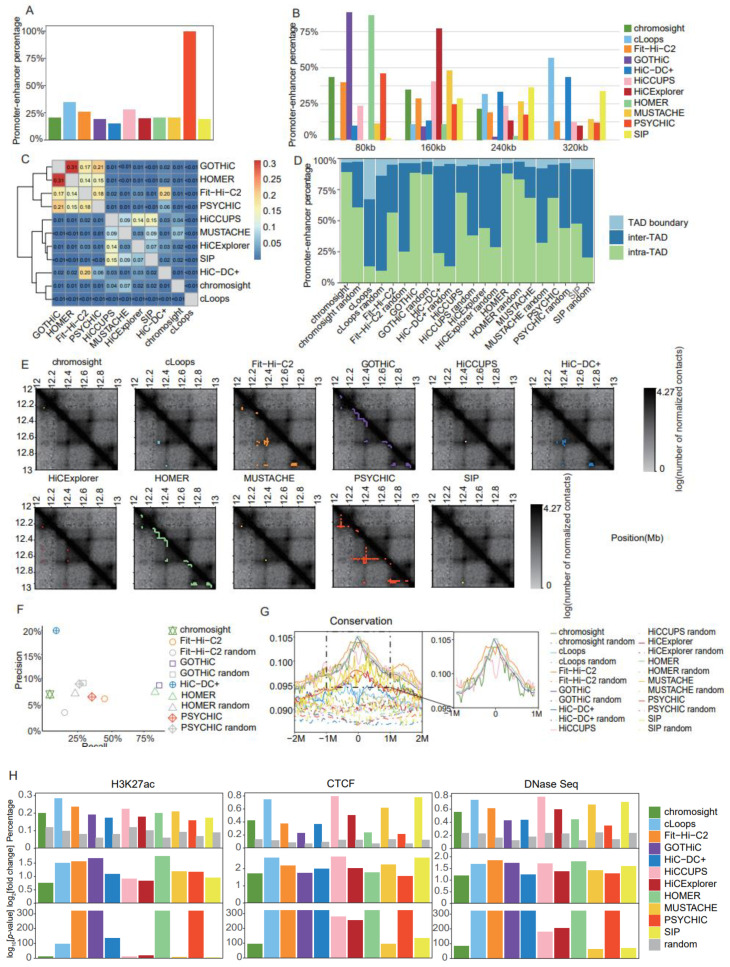
Comparison of promoter–enhancer interaction identification tools in Rao’s K562 dataset at 10 kb resolution. (**A**) Barplot of the percentage of promoter–enhancer interactions in chromatin interactions. (**B**) Barplot of distance distribution between promoters and enhancers in 80 kb intervals, at 0 kb, 80 kb, 160 kb, and 240 kb. (**C**) Heatmap for the Jaccard Indices of promoter–enhancer interactions between distinct tools. (**D**) Barplot for proportions of the two promoter–enhancer interaction anchors located in inter-TAD, intra-TAD, or TAD boundary regions, using TADs identified using HiCExplorer (25 kb and full). (**E**) Heatmaps for the contact matrices (chr18:12,000,000–13,000,000) at 10 kb resolution. Identified interactions are marked in different colors, which correspond to the various tools. (**F**) Precision–recall plot for promoter–enhancer interactions identified with each tool, and random promoter–enhancer interactions, using a CRISPR dataset as ground truth (tools for which recall and precision both equaled 0 are not shown on the graph). (**G**) Enrichment of conservation in a 4 Mb window (± 2 Mb) around the putative enhancers identified for promoter–enhancer interactions. (**H**) Barplots of the proportion of putative enhancers identified for promoter–enhancer interactions and random promoter–enhancer interactions that overlap peaks from three datasets (H3K27ac ChIP-Seq, CTCF ChIP-Seq, and DNaseSeq datasets). log_2_[fold change] and -log_10_ [*p*-value] were estimated for each tool by comparison with random promoter–enhancer interactions.

**Figure 6 ijms-24-13814-f006:**
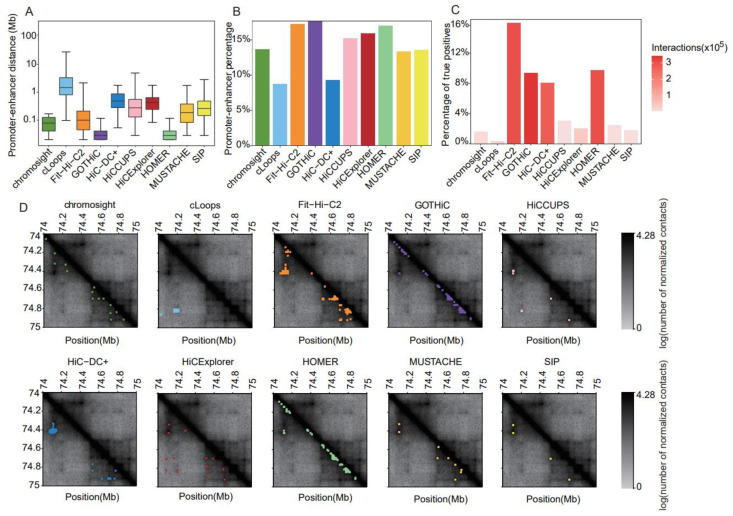
Comparative results of methods for identifying chromatin interactions using Rao’s GM12878 dataset at 5 kb resolution. (**A**) Boxplot of distances between two anchors of chromatin interactions. (**B**) Barplot of the percentage of promoter–enhancer interactions in chromatin interactions. (**C**) Barplot of the proportion of true-positive interactions recalled by the identified chromatin interactions. (**D**) Heatmaps of the contact matrices (chr18:74,000,000–75,000,000) at 5 kb resolution. Identified interactions are marked in different colors, which correspond to the various tools.

**Figure 7 ijms-24-13814-f007:**
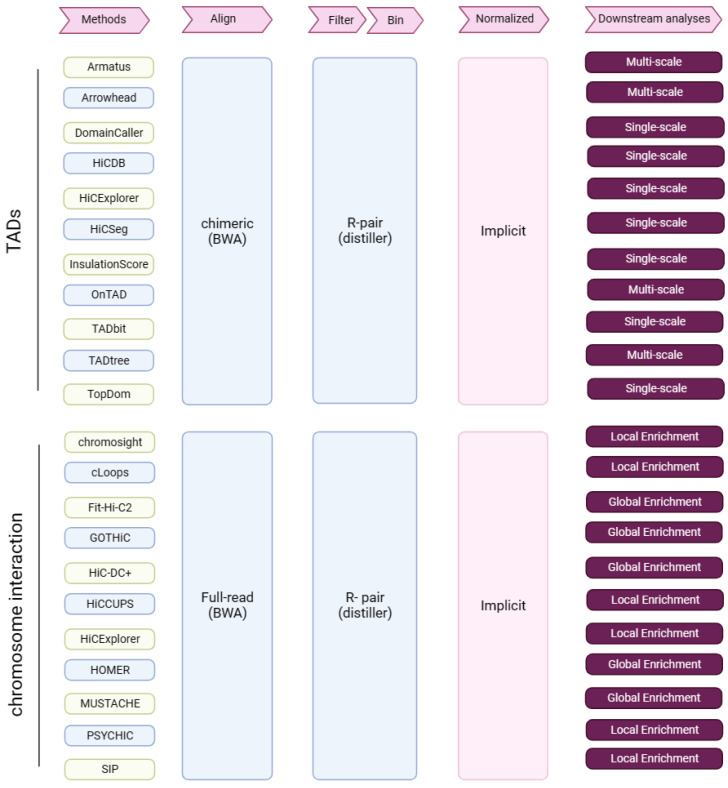
Research summary. Tools for the identification of TADs and chromatin interactions, and from Hi-C data through key analysis steps (pink arrows). R-pair—read pair-level filtering.

**Table 1 ijms-24-13814-t001:** Ratio of aligned read pairs in GM12878.

	Method	Percentage of Aligned Read Pairs (%)	Percentage of Valid Mapped Reads (%)
bwa	distiller-nf	87.21	72.37
	Juicer	94.15	71.58
Bowtie2	HiC-Pro	89.18	58.38
	HiCExplorer	62.57	40.34
	HOMER	73.29	60.96
GEM	TADbit	80.67	71.35

**Table 2 ijms-24-13814-t002:** Details of the samples used in this study.

Cell Type	Restriction Enzyme	SRA Accession Number
GM12878	MboI	SRR1658572
GM12878	MboI	SRR1658592
GM12878	MboI	SRR1658593
GM12879	MboI	SRR1658594, SRR1658595
GM12880	MboI	SRR1658596, SRR1658597
GM12878	MboI	SRR1658598
GM12878	MboI	SRR1658599
GM12878	MboI	SRR1658600
GM12878	MboI	SRR1658601
GM12878	MboI	SRR1658602
GM12878	MboI	SRR1658603
K562 (CCL-243)	MboI	SRR1658693
K562 (CCL-243)	MboI	SRR1658694
K562 (CCL-243)	MboI	SRR1658695
K562 (CCL-243)	MboI	SRR1658696
K562 (CCL-243)	MboI	SRR1658697
K562 (CCL-243)	MboI	SRR1658698
K562 (CCL-243)	MboI	SRR1658699
K562 (CCL-243)	MboI	SRR1658700
K562 (CCL-243)	MboI	SRR1658701
K562 (CCL-243)	MboI	SRR1658702

## Data Availability

All of the tools that we tested are open access and available through the links indicated in the corresponding publications. The publicly available Hi-C datasets used for the research presented in this manuscript were downloaded from the GEO (GSE63525). The ChIP-seq and DNase-seq datasets used for this research were downloaded from GEO (GSE30263, GSE51334, GSE70482, GSE25344, GSE107726).

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
