# Peer review of "Revisiting Assessment of Computational Methods for Hi-C Data Analysis"

_ijms, 2023, doi:10.3390/ijms241813814_

Round 1
Reviewer 1 Report
HiC data is a very popular source of data of chromatin conformation. Since early work, it has been observed that data have some structure, such as topologically associated domains (TADs), loops, and other features. Consequently, many tools have emerged to automatically search for these features. The present work is devoted to a comparative analysis of HiC data processing and analysis tools. Comparative analysis of tools is very important for users. This article is an original article, considers not only old approaches but also the latest tools. I believe the material is sufficiently comprehensive and does not require additional controls. The conclusions follow entirely from the main body of the paper.
The list of References is quite comprehensive (39) for this kind of article. It should be mentioned the work of Aljogol D, Thompson IR, Osborne CS, Mifsud B. Comparison of Capture Hi-C Analytical Pipelines. Front Genet. 2022 Jan 28;13:786501. doi: 10.3389/fgene.2022.786501. PMID: 35198004; PMCID: PMC8859814.
There is a small wish on the work. I would like to see the results of comparative analysis not only in the form of text and figures, but also in tables. It seems to me that such presentation of information would be more convenient for perception.
Reviewer 2 Report
Following are my critiques/notes for the manuscript:
1. Although many of the methods used to evaluate the algorithms such as HicRep are well known in the field, a short description of the exact purpose of the tool in the results section will help any uninitiated reader understand their use. I suggest including these in results section wherever it is appropriate.
2. Similarly including full form of an acronym when it first appears in the manuscript is important. MoC/JI may be familiar to some but not all readers. These terms are first used as acronym and only elaborated in methods section.
3. For methods that produce nested TADs it may not be fair to compare them with other methods. For example what level of TADs were used for the analyses of CTCF enrichment, inter v. intra TAD interaction enrichment analyses. The authors should include analyses with various levels of the TADs called by nested methods to make this comparison exhaustive.
4. Figure 4G needs to be re-factored as it is showing too much information that is not easy to parse.
5. Figure 4H are the P values -log10? not sure if this is mentioned anywhere
No comments
Reviewer 3 Report
The reviewed manuscript presents the research results regarding applying various state-of-the-art techniques for the complete end-to-end pipeline of Hi-C data analysis. The topic is interesting and actual since Hi-C data analysis can allow us to improve our understanding of genome organization and its role in cellular processes. The manuscript is well structured, and the abstract briefly reflects the manuscript's content. The results ate interesting. In my view, the main parts of the manuscript are also presented correctly. However, I think it is necessary to make some corrections before the manuscript is accepted. Bellow, I present my remarks.
1. To improve the manuscript's readability, it will be better to allocate the Related Work section, which to analyze the current works in this subject area with the allocation of both disadvantages and shortcomings of the appropriate methods. At the end of this section, allocating the unsolved parts of the general problem will be better.
2. in this case, the Introduction section should contain the Importance of the solved problems, the existing ways of this problem solution, limitations of the current methods and the main contributions of the authors' research.
3. I understand that there are various allowed structures of the manuscripts. However, my experience as both the reviewer and author of the MDPI edition's journals indicates that it will be better to present the theoretical part before the experimental one. Thus, after The Related Work section, it will be better to present the Materials and Methods. In this case, the manuscript will be more readable if you present the block chart of the research stepwise procedure and used methods. Then, you can present the results and their discussion. I think that, in this instance, the manuscript will be more readable.
4. It is necessary to add the conclusion section, which shortly describes obtained results, their advantages and limitations and further perspectives of the authors' research.
Round 2
Reviewer 3 Report
Thanks, I have no other questions.